# The Luria-Nebraska Neuropsychological Battery Neuromotor Tasks: From Conventional to Image-Derived Measures

**DOI:** 10.3390/brainsci12060757

**Published:** 2022-06-08

**Authors:** Daniele Corbo, Donatella Placidi, Roberto Gasparotti, Robert Wright, Donald R. Smith, Roberto G. Lucchini, Megan K. Horton, Elena Colicino

**Affiliations:** 1Neuroradiology Unit, Department of Medical and Surgical Specialties, Radiological Sciences and Public Health, University of Brescia and ASST Spedali Civili Hospital, 25121 Brescia, Italy; roberto.gasparotti@unibs.it; 2Occupational Medicine, Department of Medical and Surgical Specialties, Radiological Sciences and Public Health, University of Brescia, 25121 Brescia, Italy; donatella.placidi@unibs.it (D.P.); rlucchin@fiu.edu (R.G.L.); 3Department of Environmental Medicine and Public Health, Icahn School of Medicine at Mount Sinai, New York, NY 10029, USA; robert.wright@mssm.edu (R.W.); megan.horton@mssm.edu (M.K.H.); elena.colicino@mssm.edu (E.C.); 4Department of Microbiology and Environmental Toxicology, University of California, Santa Cruz, CA 95064, USA; drsmith@ucsc.edu; 5Environmental Health Sciences, School of Public Health, Florida International University, Miami, FL 33199, USA

**Keywords:** Luria neuromotor test, sensorimotor impairments, image-derived scoring

## Abstract

Background: Sensorimotor difficulties significantly interfere with daily activities, and when undiagnosed in early life, they may increase the risk of later life cognitive and mental health disorders. Subtests from the Luria-Nebraska Neuropsychological Battery (LNNB) discriminate sensorimotor impairments predictive of sensorimotor dysfunction. However, scoring the LNNB sensorimotor assessment is highly subjective and time consuming, impeding the use of this task in epidemiologic studies. Aim: To train and validate a novel automated and image-derived scoring approach to the LNNB neuro-motor tasks for use in adolescents and young adults. Methods: We selected 46 adolescents (19.6 +/− 2.3 years, 48% male) enrolled in the prospective Public Health Impact of Metal Exposure (PHIME) study. We visually recorded the administration of five conventional sensorimotor LNNB tasks and developed automated scoring alternatives using a novel mathematical approach combining optic flow fields from recorded image sequences on a frame-by-frame basis. We then compared the conventional and image-derived LNNB task scores using Pearson’s correlations. Finally, we provided the accuracy of the novel scoring approach with Receiver Operating Characteristic (ROC) curves and the area under the ROC curves (AUC). Results: Image-derived LNNB task scores strongly correlated with conventional scores, which were assessed and confirmed by multiple administrators to limit subjectivity (Pearson’s correlation ≥ 0.70). The novel image-derived scoring approach discriminated participants with low motility (<mean population levels) with a specificity ranging from 70% to 83%, with 70% sensitivity. Conclusions: The novel image-derived LNNB task scores may contribute to the timely assessment of sensorimotor abilities and delays, and may also be effectively used in telemedicine.

## 1. Introduction

Sensorimotor dysfunction is a pervasive developmental disorder with a prevalence of up to 19% among school-aged children and adolescents [1,2]. Sensorimotor difficulties significantly interfere with daily activities and academic performance, and, if untreated, may lead to an increased risk of cognitive and mental health disorders later in life [1,2]. Childhood sensorimotor dysfunction is defined by decrements in fine and/or gross motor skills, with poor motor performance that is usually slower, less accurate, and more variable than that of an unaffected individual [3]. It is also associated with reduced learning ability, verbal and visuospatial memory problems, and cognitive impairments [1,2,4]. Sensorimotor dysfunction is often a comorbidity with other developmental disorders, such as attention deficit and hyperactivity disorder. Cognitive impairments in children with sensorimotor conditions persist into adolescence and adulthood, and may even extend beyond their motor difficulties [1,5]. Due to the common root and interactions between sensorimotor and cognitive disorders, sensorimotor dysfunction is considered a condition of the central nervous system [1,2,6]. Early detection of abnormal sensorimotor function can lead to timely interventions able to mitigate the adverse long-term consequences [5,6].

Several different tests and exams are designed to detect childhood and adolescence sensorimotor impairment [7,8,9]. The Luria-Nebraska Neuropsychological Battery (LNNB) stands out for its evaluation of the presence of neuro-motor impairments and identification of focal brain abnormalities that may account for these impairments in children and adolescents [10].

The standard LNNB consists of 11 clinical scales assessing major areas of neuropsychological functioning (motor functions, rhythm, tactile functions, visual functions, receptive speech, expressive speech, writing, reading, arithmetic, memory, intellectual processes), two sensorimotor scales (left hemisphere, right hemisphere), and three summary scales (pathognomonic, profile elevation, impairment) [10]. Although LNNB can capture early sensorimotor impairments, it is not widely used in epidemiological and clinical studies largely because it is both highly subjective and time consuming [10,11,12]. Indeed, while the administration of the LNNB is standardized, the scoring and thus the interpretation of the LNNB data are time-consuming duties and require the judgment of a trained administrator who is responsible for evaluating the appropriateness of the movement and counting the participant’s movements during a specific time frame [10,11,12].

To address the subjectivity and timeliness of the LNNB in epidemiologic studies, we developed an automated scoring algorithm using a novel mathematical approach. We applied this approach to LNNB data collected from a subset of adolescents enrolled in the ongoing Public Health Impact of Metal Exposure (PHIME) study. Specifically, we trained and validated five image-derived sensorimotor LNNB task scores able to assess the adolescent’s movement by combining optic flow fields from recorded image sequences on a frame-by-frame basis. We then validated the accuracy of those novel image-derived LNNB task scores comparing them with conventional LNNB scores. To limit the subjectivity of the conventional LNNB scores, multiple administrators evaluated the appropriateness and counted the participant’s movements. This novel automatic image-derived sensorimotor assessment may be implemented during telemedicine services, thus leading to more personalized treatments and earlier interventions in subjects at a higher risk of developing sensorimotor impairment.

## 2. Methods

**Study Population:** To test our novel scoring algorithm, we selected 46 participants (ages 15–23; female = 52%) from the ongoing PHIME cohort consisting of 208 children and adolescents, who were enrolled between 2007 and 2014, and followed up between 2017 and 2021. The overall objective of the study was to examine associations between metal exposure and neurodevelopmental outcomes [11]. Participants were at first identified through the local school districts and enrolled with the following inclusion criteria: born in the respective area to a family who resided in the area for at least a generation, lived in the study area since birth [11]. Exclusion criteria included: known hand or finger motor deficits, visual deficits not adequately corrected, and any history of neurological, metabolic, hepatic, or endocrine diseases [11]. Participants were also excluded if they had a history of receiving parenteral nutrition that may cause overload of environmental chemicals (i.e., manganese), or they were taking prescription psychoactive drugs or had known psychiatric disturbances [11]. All participants were followed over adolescence and, during the follow-up visit, 46 of them underwent a LNNB test [11]. The 46 participants were randomly selected and received a sensorimotor LNNB evaluation from two trained administrators, who assessed the appropriateness and counted the participant’s movements. Both administrators agreed on the correctness and the number of participant’s movements.

**Ethics:** Written informed consent was obtained from parents and children. Study protocols were approved by the institutional review board at the Ethical Committee of the Public Health Agency of Brescia and Mount Sinai. Extensive data on exposure assessment in these areas were published previously [11].

**The Luria-Nebraska Neuropsychological Battery:** The LNNB is a battery of motor-neuron tests used in combination with other neurodevelopmental exams to identify childhood or adolescence motor-cognitive deficits or delays [10,11,12]. Subjects were tested with the LNNB according to the instructions provided in the test manual [13]. Briefly, participants were told to complete the movement/task for 10 s. This motor coordination exam consisted of a standardized test battery of five tasks [10,11]. The five movements/tasks include the dominant hand clench (Task 1), the finger–thumb touching with the dominant hand (Task 2), the non-dominant hand clench (Task 3), the finger–thumb touching with non-dominant hand (Task 4), and alternative hand clench (Task 5) [11]. The sum of the frequency of each of the five task yields a final score that reflects a motor score [11]. The test administrator kept track of time and the frequency of tasks completed in the 10 s period.

**Analytical plan:** Our novel scoring method leverages the image-derived LNNB task scores with the following steps:

*Visual record of all actions.* A trained administrator visually recorded the participant’s hand movements via an iPad Pro camera (Camera module iSight with8 megapixel) and stored the video in a local hard drive. All participants were seated at a determined distance (1.5 m) from the camera.

*Normalization.* To remove variability given by sex, age, and recording time, the hand movements in each image were normalized to a pre-defined number of voxels, resolution, and duration using MATLAB scripts.

*Compute optic flow.* The local motion vectors or optic flow fields were identified from an image sequence on a frame-by-frame basis. Optical flow is an image analysis technique used to detect motion in video sequences [14]. In each image sequence, the optical flow generates a vector for each image pixel representing the apparent motion in the corresponding sampling period. The movement represented by the optical flow is considered an apparent movement because it is a projection of the real three-dimensional (3D) image on a two-dimensional (2D) plane. In addition, previous optical flow computation techniques analyze the brightness variations of each pixel, thus making it impossible to distinguish between true and apparent motion.

We computed the local motion vector for each pixel in the image on a frame-by-frame basis as previously done [15]. The instantaneous full velocity at x, y, z pixel locations that results from a 3D camera rotation, ω = (ω_x_, ω_y_, ω_z_)^T^ with ω_p_ the angular velocity around the p-axis (with p = x, y and z), can be well approximated by v(x) = B(x)ω [16], where the B(x) matrix can be defined as:$$B_{\left(x\right)}=\left[\begin{array}{ccc} \mathrm{xy}/f & \left(-f-x^{2}\right)/f & y\\ \left(f+y^{2}\right)/f & -\mathrm{xy}/f & -x \end{array}\right]$$
where f is the focal length of the camera.

The quality of the velocity estimates can be greatly influenced by camera shocks and vibrations. Using this algorithm, we removed this unstable component of the camera motion and stabilized the image sequence, maximizing the temporal local velocity constancy over the entire short sequence. This method, which is embedded within a phase-based optic flow algorithm, was previously tested on both synthetic and complex real-world sequences [17]. The temporal evolution of contours of constant phase can yield a good approximation to the local velocity field [18]. We used this approach because of its computational efficiency as it involves linear systems and simple transformations, the result of which can be computed without time-consuming re-filtering.

*Motion quality*. We estimated the motional quality of each image-derived LNNB task score as the mean motion over the whole recorded video, as previously done [19]. We analyzed the local motion energy (i.e., speed) of the participant’s hand movements in each recorded video as a function of time by averaging speed over pixels [19]. We then calculated mean local motion speed temporal profiles. This resulting measure was expressed in degree/s and provided information about the quantity of motion performed by the hand of the tested subject [19].

*Correlation between conventional and image-derived LNNB scores and accuracy evaluation of the image-derived LNNB scores.* We estimated the association between conventional and image-derived LNNB task scores with Pearson correlation coefficients. To determine the accuracy of the image-derived LNNB score, we first classified participants with low motility using both conventional and image-derived LNNB scores, and we considered low motility when participant’s LNNB scores were below their corresponding mean population levels. We then evaluated the accuracy of the image-derived LNNB scores using the Receiver Operating Characteristic (ROC) curve and the area under the ROC curve (AUC).

*Code availability:* MATLAB code is publicly available on https://github.com/danielecorbo/Luria/blob/c8cf999e92f5367b60d69e0a6915ca5347c43d6e/motion_calculation.m (accessed on 14 April 2022).

## 3. Results

PHIME population description. PHIME participants were 19.6 (+/−2.3) years of age and were equally divided into males (48%) and females (52%). The majority of adolescents did not smoke (67%) and did not report any alcohol consumption (57%) (Table 1). All completed the five LNNB tasks (Table 2). The tasks performed by the non-dominant hand (i.e., Task 2 and Task 5) showed lower scores (lower motility) and higher variability (less precision in the movement) compared to the same tasks with the dominant hand (Task 1 and Task 4, respectively) (Table 2).

**The image-derived LNNB task scores.** Visual recordings of the five sensorimotor LNNB tasks completed by the 46 participants led to 240 visual recordings used to create the novel image-derived LNNB task scores. From each video, we obtained: (a) conventional LNNB task scores from two trained administrators, who both counted the number of performed actions by visual inspection, and agreed on the correctness of the participants’ movements and the number of actions; and (b) the novel predicted image-derived LNNB task scores computed via MATLAB. Distributions and descriptive statistics of both conventional and image-derived LNNB tasks are provided in Table 2 and Appendix A.

We then evaluated the correlation between conventional and image-derived LNNB task scores. All tasks showed a strong linear relationship between conventional and image-derived LNNB task scores (Figure 1, Appendix A). Specifically, the Pearson correlation coefficients between the conventional and image-derived LNNB task scores ranged between 0.70 and 0.74, with all coefficients statistically significant different from 0 (Figure 1, Appendix A).

In this subset and using the conventional LNNB task scores (from Task 1 to Task 5, respectively), 25, 26, 27, 22, and 26 participants showed low motility, i.e., their conventional LNNB scores were below the corresponding mean population levels. With the novel estimated image-derived LNNB tasks and setting a sensitivity level of 70%, we discriminated participants with low levels of motility with a specificity of 71%, 83%, 71%, 63%, 67% for Task 1 to Task 5, respectively. The accuracy of those models ranged from 70% for Task 4 to 83% for Task 2 (Figure 2, Appendix A).

## 4. Discussion

This is the first study to develop automatic and image-derived LNNB task scores in adolescents using a novel mathematical approach combining optic flow fields from image sequences on a frame-by-frame basis. This novel image-derived LNNB task-scoring approach accurately estimated an alternative for conventional LNNB measurements and provided good discrimination of participants with lower scores, indicating motility below the mean population values. These findings supported the hypothesis that imaging data can lead to novel automatic tools able to provide a timely sensorimotor screening and discriminate individuals at higher risk of developing sensorimotor impairments and delays.

Our findings were consistent with prior literature showing that the dominant hand is faster [20], more accurate [21], and less variable [20] in movements than the non-dominant hand. This supports the consensus regarding the specialization of the non-dominant system for utilizing proprioceptive feedback [22,23,24], defined as the central motor ability to sense the position and the movement of a limb in space along with muscular effort and tension [25]. The proprioceptive feedback is also strongly connected with peripheral or central nervous structures [25]. Our findings also showed good accuracy (≥70%) in the estimation of LNNB task score alternatives using image-derived data, especially in evaluating the non-dominant hand clench tasks. This can be explained by the fact that the non-dominant hand provides more accurate information on proprioception [25]. Indeed, this information cannot be compensated for by the advantages of the dominant hand, such as increased training of the hand muscles [26], the enlarged excitability of the dominant motor cortex [27], and the increased excitability of motor-neuronal pool at the level of spinal circuitry [28]. The conventional and image-derived LNNB scores also showed strong correlation coefficients (≥0.70).

To determine the image-derived LNNB task scores, we employed a novel mathematical approach leveraging the image sequence on a frame-by-frame basis and the local motion energy as a function of time. The contribution of this approach was two-fold. First, we used a normalized, robust, and computationally efficient method, which facilitates the detection of movements that otherwise may be discarded or overestimated by the visual count. Second, our strategy may limit time-consuming duties for the trained administrator and may improve the timeliness of the assessment of sensory-motor impairments or delays. Indeed, this novel LNNB scoring approach did not require any administrator and it may be considered as sensorimotor screening during telemedicine services. We then compared our image-derived LNNB task scores with conventional scores, which are highly subjective. In this study, we were able to limit the subjectivity of conventional LNNB task scores with the assessment by multiple trained administrators. This allowed us to properly characterize the adolescents’ movement and to correctly estimate the number of actions performed. We finally included the code to facilitate reproducibility and transparency of our scientific results.

Our study population consisted of healthy adolescents, thus limiting the generalizability of our findings to populations of other ages and different sensorimotor abilities. However, these preliminary results showed the potential of recorded videos for telemedicine applications that aim to provide early diagnostics of any sensorimotor impairments and delays, including Parkinsonians-like symptoms. An additional limitation was the relatively small sample size, which led us to internally validate our results and to potentially overfit our findings. To overcome these limitations and improve the performance of these novel image-derived LNNB task-scoring approach, further studies should include larger populations of different ages and sensorimotor abilities.

## 5. Conclusions

We trained and validated an automatic image-derived LNNB task-scoring approach able to assess an individual’s movements. This novel image-derived LNNB task-scoring approach may mitigate administrator’s subjectivity, limit time-consuming duties, and provide the groundwork for early diagnostics for any sensorimotor impairments and delays.

## Figures and Tables

**Figure 1 brainsci-12-00757-f001:**
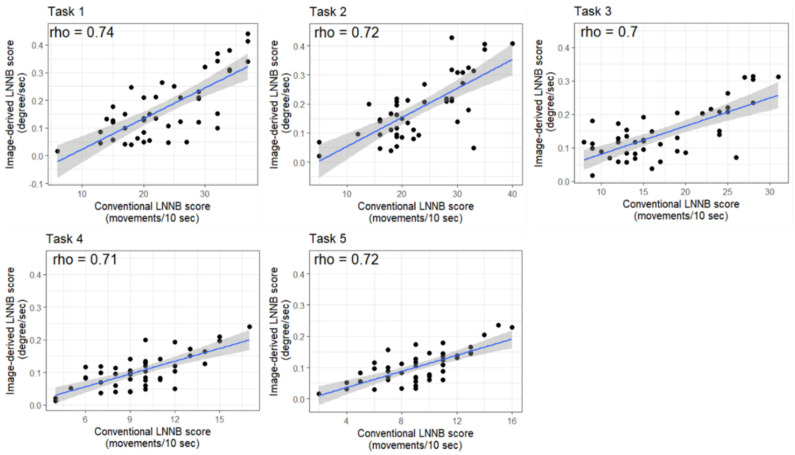
Scatterplots, Pearson correlation coefficients (rho) and linear trends of the relationship between the conventional (x-axis) and the image-derived (y-axis) Luria-Nebraska Neuropsychological Battery (LNNB) task scores. Each dot represents an individual, the black line represents the linear trend of the association between conventional and the image-derived LNNB task scores, and the grey shade indicates the 95% Confidence Interval of the linear trend. Task 1: the dominant hand clench, Task2: the finger–thumb touching with the dominant hand, Task 3: the non-dominant hand clench, Task 4: the finger–thumb touching with non-dominant hand, Task 5 alternative hand clench.

**Figure 2 brainsci-12-00757-f002:**
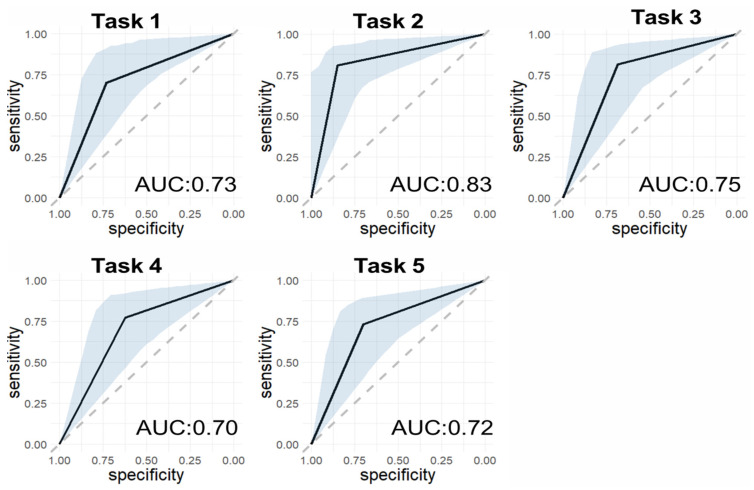
Receiver Operating Characteristic (ROC) curve (solid black line), empirical 95% confidence interval (grey shade), and area under the ROC curve (AUC) of the image-derived Luria-Nebraska Neuropsychological Battery (LNNB) tasks, classifying participants with lower mobility (LNNB score < population mean levels). Dashed grey line indicates the line of no-discrimination. Task 1: the dominant hand clench, Task2: the finger–thumb touching with the dominant hand, Task 3: the non-dominant hand clench, Task 4: the finger–thumb touching with non-dominant hand, Task 5 alternative hand clench.

**Table 1 brainsci-12-00757-t001:** Sociodemographic characteristics of selected participants from the Public Health Impact of Metal Exposure (PHIME) Study.

Characteristics	N (%) or
	Mean (SD)
Age (years)	19.6 (2.3)
Sex (male)	22 (48%)
Maternal education	
≤High School	38 (83%)
>High School	3 (6%)
NA	5 (11%)
Self-reported cigarette smoking	
No	31 (67%)
Yes	7 (15%)
NA	8 (17%)
Self-reported alcohol consumption	
No	26 (57%)
Yes	12 (26%)
NA	8 (17%)

%: Percentage, SD: Standard Deviation; NA: Not Available.

**Table 2 brainsci-12-00757-t002:** Conventional and image-derived Luria-Nebraska Neuropsychological Battery (LNNB) test scores among 46 PHIME participants.

Conventional	Mean	Standard Deviation	CV	1st Quartile	2nd Quartile	3rd Quartile	Minimum	Maximum
Task 1	23.48	7.54	2.42	18	22	29	6	37
Task 2	23.15	7.69	2.56	19	21.5	29	5	40
Task 3	17.5	6.31	2.28	13	15.5	23.75	8	31
Task 4	9.67	2.86	0.85	8	10	11	4	17
Task 5	9.13	2.96	0.96	7	9	11	2	16
**Image-derived**	**Mean**	**Standard Deviation**	**CV**	**1st Quartile**	**2nd Quartile**	**3rd Quartile**	**Minimum**	**Maximum**
Task 1	0.17	0.11	0.07	0.08	0.13	0.24	0.02	0.44
Task 2	0.18	0.11	0.06	0.1	0.17	0.22	0.02	0.43
Task 3	0.14	0.08	0.04	0.09	0.12	0.2	0.02	0.31
Task 4	0.1	0.05	0.03	0.07	0.1	0.13	0.01	0.24
Task 5	0.1	0.05	0.03	0.06	0.09	0.14	0.02	0.24

CV: Coefficient of Variation. Task 1: the dominant hand clench, Task 2: the finger–thumb touching with the dominant hand, Task 3: the non-dominant hand clench, Task 4: the finger–thumb touching with non-dominant hand, Task 5: alternative hand clench.

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
