# Peer review of "The Luria-Nebraska Neuropsychological Battery Neuromotor Tasks: From Conventional to Image-Derived Measures"

_brainsci, 2022, doi:10.3390/brainsci12060757_

Round 1

Reviewer 1 Report

Thank you for the opportunity to review this research/manuscript.  I found the study fascinating and believe it's important, especially when it comes to speeding the diagnosis of sensorimotor dysfunction (at least in adolescents).  I hope this article will motivate other researchers to expand on this research (e.g. larger sample size, different populations).

Author Response

We thank the reviewer for the comment and appreciation. In the future we will try to increase the sample and differentiate the population studied.

Reviewer 2 Report

It is interesting work concerning the accuracy of the novel automatic image-derived LNNB task scoring approach based on Receiver Operating Characteristic (ROC) curves and Areas under ROC curves (AUC). The results revealed that the novel image-derived scoring approach discriminated participants with low motility with a specificity ranging from 70% to 83%, with 70% sensitivity indicating the potential for use in telemedicine. The methods and statistical analysis are appropriate.

I have several comments:

The authors could add information on the criteria according to which sensitivity and specificity were determined. In other words, is the sensitivity and specificity of the novel LNNB method sufficient?

To what extent can gender differences play a role?

Author Response

We thank the reviewer for this thoughtful comment. We agree with the reviewer that further heterogeneous studies with larger sample size should additionally evaluate the sensitivity and specificity of our approach. To facilitate applications and replications in those further studies we made the code available on our repository and we included the link in the manuscript.

In addition we normalized the frame-by-frame images by sex to mitigate the heterogeneity given by sex on our initial values. Further studies should also evaluate whether we have an additional sex-effect in our analysis. We discussed all these points in the discussion of the manuscript.